# Thermal–Fluid–Solid Coupling Analysis on the Effect of Cooling Gas Temperature on the Fatigue Life of Turbine Blades with TBCs

Yingtao Chen [1], Ziliang Zhang [1,*], Yanting Ai [1], Peng Guan [1], Yudong Yao [2] and Hongwei Liu [1]

[1] School of Aero Engine, Shenyang Aerospace University, Shenyang 110136, China; chenyingtao@163.com (Y.C.); ytai@163.com (Y.A.); 613511gp@163.com (P.G.); syhkhtdxlhw@163.com (H.L.)
[2] School of Power and Energy, Northwestern Polytechnical University, Xi'an 710072, China; yaoyudong148@163.com
* Correspondence: zzlari@163.com

**Abstract:** The application of thermal barrier coatings (TBCs) can increase the blade's operating temperature and fatigue life. Previous studies have neglected the effects of cooling gas temperature variations on the temperature field, stress field, and fatigue life of blades and TBCs. For this reason, this paper considers the inhomogeneity of the high-temperature gas inlet temperature, the internal cooling gas temperature, the periodicity of the external flow field, etc., and establishes a finite element model of the gas turbine blade with TBCs. Then, the life of the blade and the TBCs is predicted based on the Ncode 2020R2. Finally, the effect of the cooling gas temperature on the temperature, stresses, and life of the blade and the TBCs is analyzed. The results show that the fatigue life of the TBCs is lower than that of the blade, and the low-life region of the TBCs is located at the leading edge of the blade, which is consistent with the TBCs shedding region of the real blade and verifies the accuracy of the life prediction method in this paper. The fatigue life of the blade and TBCs firstly increases and then decreases with the increase in the cooling gas temperature, and the trend of the stress changes in the opposite direction. When the cooling gas temperature is increased from 573 K to 873 K, the minimum life of the blade is increased by 358.5%, and that of the TBCs is increased by 138.7%. The conclusions can provide guidance for the design of long-life turbine blades with TBCs.

**Keywords:** turbine blade; TBCs; thermal–fluid–solid coupling method; fatigue life; cooling gas temperature; thermal stress

## 1. Introduction

The aero-engine is the "heart" of the aircraft; it directly affects the reliability and safety of the aircraft flight process [1,2]. Turbine blades, as the core components of gas turbine engines, play a key role in the operation of aero engines. With the increasing thrust-to-weight ratio of aero-engines, turbine blades need to withstand higher service temperatures, and turbine inlet temperatures have far exceeded the melting point of metallic materials [3,4]. TBCs are a kind of low thermal conductivity, high-temperature resistant material. TBCs coated on the blade's surface can effectively reduce the alloy's surface temperature and improve the blade's operating temperature and fatigue life, which is one of the effective methods to meet the turbine blade's long service time [5–7].

Several research scholars have conducted systematic studies on turbine blades not covered with TBCs. Methods for analyzing turbine blades' heat transfer, stress, and thermal fatigue life have matured. There are more mature programs for fatigue life research in blade temperature field simulation and stress field simulation. Peng et al. [8] conducted thermal shock fatigue tests on blades in the last century. The results show that thermal stresses tend to induce cracks in regions with large temperature differences, such as at the hole edges. Rezazadeh et al. [9] calculated and analyzed the blade temperature distribution

and fatigue life using the conjugate heat transfer (CHT) method. It was found that the point of risk of blade fatigue failure (maximum equivalent stress) is located in the fir-tree region of the blade. Javad et al. [10] used the modified conjugate heat transfer (CHT) method to numerically analyze the structure and thermal stresses of the first-stage turbine blade. The results show that temperature-dependent materials are more accurate in predicting the fatigue life of turbine blades. Cai et al. [11] used the thermal–fluid–solid coupling method to calculate the blade temperature field and stress field. It is concluded that the thermal–fluid–solid coupling method can effectively predict the thermal stress damage of the blade; the thermal stress on the blade surface has a greater relationship with the local temperature gradient of the blade. Guan et al. [12] combined fluid–solid coupling in the numerical analysis of turbine blades to calculate the maximum temperature, cooling efficiency, and thermal stresses of a certain type of blade. Yin et al. [13] used a coupled thermal–fluid–solid approach to analyze the temperature and stress fields of the simplified blade. The coupling method can effectively control the error of results. Qian et al. [14] estimated the service life of high-pressure turbine guide vanes based on the thermoelastic coupling method and obtained results similar to those of the tests. According to the above literature, it is found that the thermal–fluid–solid coupling method can obtain the temperature and stress fields of the blade more accurately than the CHT method. The simulation of the temperature and stress fields plays a crucial role in fatigue life prediction.

On this basis, further introduction of TBCs for research is of great practical significance to improve the service life and strength of turbine blades. When the blades with TBCs are used for a long period in a high-temperature environment due to high-temperature loading and stress concentration, the structure of TBCs undergoes significant changes, which is accompanied by the emergence and expansion of surface cracks, ultimately leading to the failure of TBCs [15–18]. Sohn et al. [19] compared the state of the high-pressure turbine blades with TBCs before and after service and found that significant sintering occurred in the TBCs of the turbine blades after service, with localized cracking of the TBCs in the tip area. Yang et al. [20] modeled the thermal barrier-coated blades and conducted an in-depth study of the TBC stress field based on finite elements. Wei et al. [21] investigated the thermal insulation effect of TBCs, which was better at the leading edge of the blade than at the trailing edge according to their simulation of the temperature field. Hao et al. [22] first investigated the residual stresses and strains after the TBCs were deposited onto the blade by the finite element method and further simulated and analyzed the temperature and stress state of the blade with TBCs under high-temperature thermal shock. Shi et al. [23] investigated the effect of cooling gas flow rate on the cooling performance of turbine blades with TBCs by a gas-thermal coupling method. The results show the cooling efficiency of the thermal barrier-coated blades increases with the increase in the cooling flow rate. Toshihiko et al. [24] simulated the temperature field distribution of a blade with TBCs by CFD calculations and a one-way coupling method. The effect of internal cooling performance on the temperature distribution of the blade was analyzed.

In previous studies, more attention has been paid to the effects of thermal shock on blades and TBCs under a single operating condition (cooling gas temperatures are fixed), but fewer studies have been conducted on the effect of cooling gas temperature change on the temperature field, stress field and fatigue life of the blade and TBCs. Therefore, based on the real thermal shock environment of blades with TBCs, this paper establishes the relevant finite element model, analyzes the temperature field and stress field of the blade and TBCs, predicts the fatigue life of the blade and TBCs, and explores the influence of the cooling gas temperature change on the stress and fatigue life of the blade with TBCs.

## 2. Geometric Models and Numerical Methods

### 2.1. Geometric Model

In this paper, only the operating conditions of individual blades are considered since the turbine blades in the engine serve in the same environment. In the modeling process, to facilitate the simulation calculations, the internal cooling channels of the blade with TBCs

are simplified without considering the effect of cooling holes on the surface of the blade. The blade with TBCs and flow field model is shown in Figure 1, and the model as a whole consists of four parts: blade, TBCs, external flow field, and internal flow field. Blade wall thickness 2.5 mm, blade height 156 mm. In this paper, the dimensionless parameter $S/C_z$ is defined to characterize the location on the outer contour of the turbine blade. As shown in Figure 2, where $C_z$ denotes the chord length of the blade, and $S$ denotes the path distance along the outer contour of the blade cross-section. The TBCs is a typical two-layer structure with a bond coat layer (BC) thickness of 100 μm and a top coat layer (TC) thickness of 350 μm.

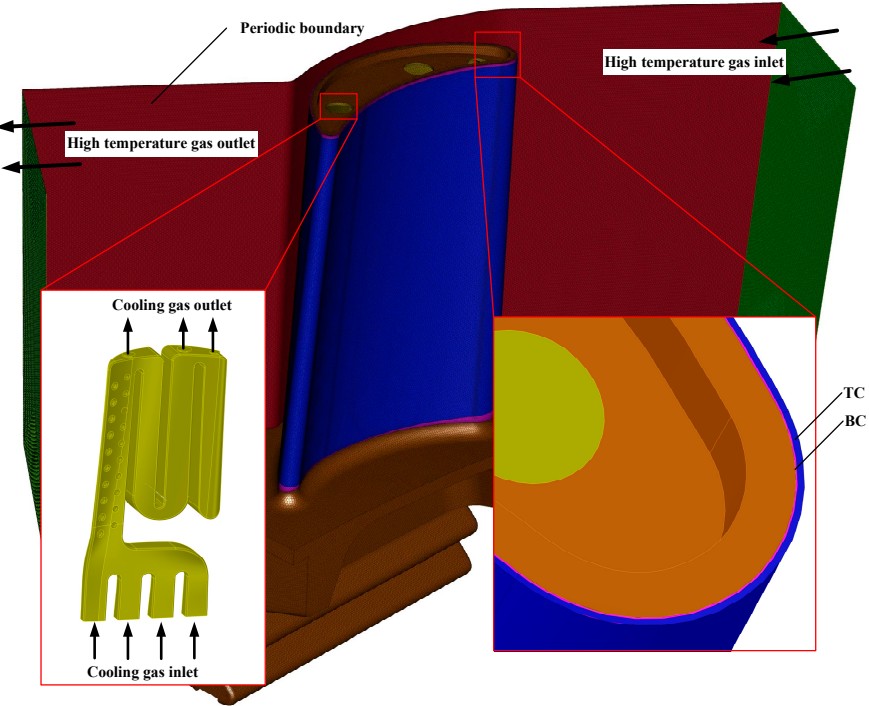

**Figure 1.** Illustration of finite element model of the blade, TBC, and flow field.

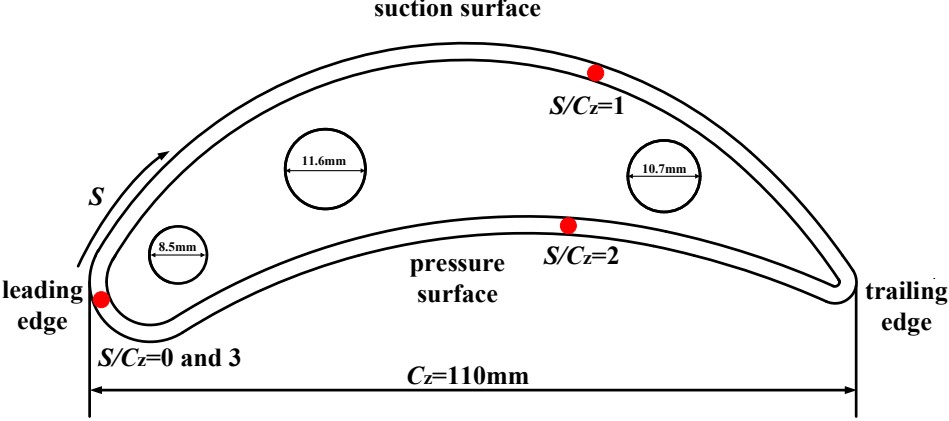

**Figure 2.** Schematic of blade top cross-section and pathway.

The fluid–solid coupling computational domain is divided into three main parts: the main flow gas domain, the blade and TBCs solid domain, and the internal cooling gas domain. Before starting the numerical study, the mesh is analyzed for independence to ensure the accuracy of the simulation. Figure 3 gives the temperature profiles of the outer profile of the blade at 50% height for a mesh number of 2.7, 6.7, and 8.7 million for the gas temperature of 1225 K and cooling gas temperature of 573 K, respectively. It can be seen

that the change in the temperature profile for a total mesh of 6.7 million almost coincides with that for a mesh of 8.7 million, and the change in the calculated results is less than 0.5%. So, this study is finally carried out numerically with a 6.7 million mesh. The mesh numbers of the dominant air domain, the blade and TBCs solids domain, and the internally cooling air domain are 2.1 million, 2.8 million, and 1.8 million, respectively. In order to more accurately reflect the action of high-temperature gas and cooling gas on the blade, a three-layer boundary layer is applied at the fluid–solid interface and the mesh in the vicinity of the boundary layer is refined to ensure that the Y+ value of the surface meets the requirements of the computational model [23].

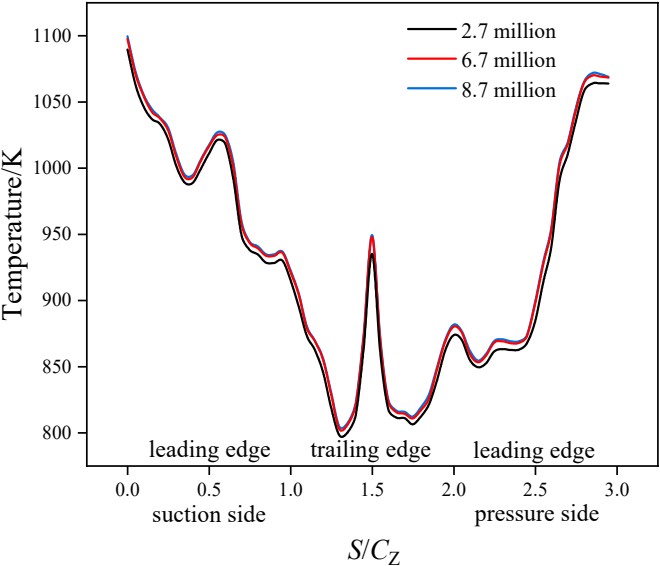

**Figure 3.** Mesh independence verification: temperature distribution.

The blade material, as well as each layer of the TBCs, including the BC layer (NiCoCrAlY) and the TC layer (8YSZ), are considered isotropic materials. The physical and thermal properties of the TBCs are shown in Table 1 [20], which are temperature-dependent and characterized by linear elasticity. The blade material is a nickel-base superalloy with a density of 8350 kg/m$^3$. The thermal conductivity, specific heat capacity, Young's modulus, Poisson's ratio, and coefficient of thermal expansion of the blade alloys at different temperatures are shown in Table 2.

**Table 1.** Temperature and material parameters of BC and TC used in FE simulations [20].

| | Temperature Range (°C) | Young's Modulus (GPa) | Poisson's Ratio | Thermal Expansion Coefficient ($10^{-6}$/°C) | Yield Strength (MPa) | Thermal Conductivity (W/(m·K)) | Specific Heat (J/(kg·K)) | Density (kg/m$^3$) |
|---|---|---|---|---|---|---|---|---|
| Bond coat | 20–1600 | 200–110 | 0.30–0.33 | 13.6–17.6 | 426–114 | 5.8–17.0 | 450 | 7380 |
| Top coat | 20–1600 | 48–22 | 0.10–0.12 | 9.0–12.2 | - | 2.0–1.7 | 505 | 3610 |

### 2.2. Boundary Conditions

In order to simulate the working environment of the blade close to the real situation, based on the experimental data, the distribution function of the total gas inlet temperature along the radial direction is obtained using the polynomial method.

$$T_g = -17524Z_r^2 + 3181.9Z_r + 1008.3 \tag{1}$$

where $T_g$ is the mainstream gas inlet temperature in K, and $Z_r$ is the radial coordinate starting at the root of the blade in m. Figure 4 shows the temperature profile of the gas inlet's total temperature along the radial direction ($H_r$ is the percentage of the radial height

of the blade). Figure 5 shows the cloud profile of the gas inlet total temperature distribution, which verifies the feasibility of the method. In order to be consistent with the actual situation, both sides of the external flow field are set as periodic boundaries.

**Table 2.** Temperature-dependent blade alloy properties and heat transfer parameters [11].

| $T_a$ K | $\lambda_a$ W/m·K | $C_{pa}$ J/kg·K | E GPa | $\mu_a$ - | $C_{te}$ $K^{-1}$ |
|---|---|---|---|---|---|
| 298.15 | 8.45 | 469 | 129.9 | 0.3 | - |
| 373.15 | 10 | 474 | 128 | 0.3 | $1.19 \times 10^{-5}$ |
| 473.15 | 11.95 | 482 | 126 | 0.3 | $1.24 \times 10^{-5}$ |
| 573.15 | 13.8 | 491 | 123 | 0.3 | $1.26 \times 10^{-5}$ |
| 673.15 | 15.5 | 501 | 118 | 0.3 | $1.29 \times 10^{-5}$ |
| 773.15 | 17.1 | 511 | 114 | 0.3 | $1.32 \times 10^{-5}$ |
| 873.15 | 18.55 | 522 | 110 | 0.3 | $1.36 \times 10^{-5}$ |
| 973.15 | 19.85 | 534 | 106 | 0.3 | $1.40 \times 10^{-5}$ |
| 1073.15 | 21 | 547 | 101 | 0.3 | $1.45 \times 10^{-5}$ |
| 1173.15 | 22 | 561 | 95 | 0.3 | $1.50 \times 10^{-5}$ |
| 1273.15 | 22.8 | 575 | 86 | 0.3 | $1.56 \times 10^{-5}$ |

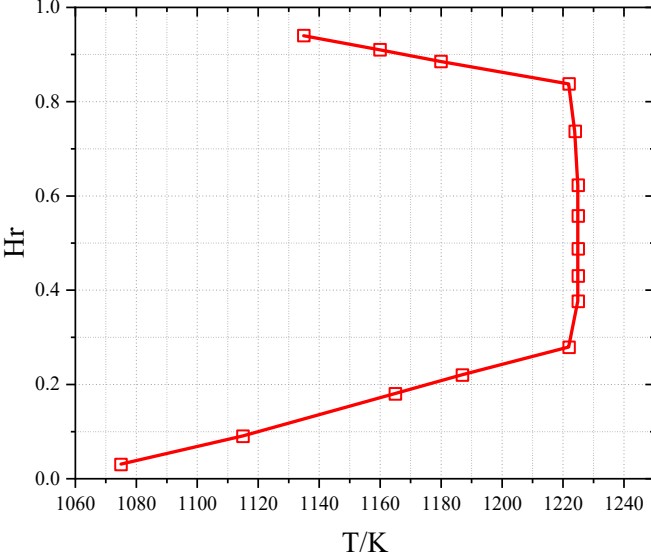

**Figure 4.** Temperature curve of mainstream gas inlet.

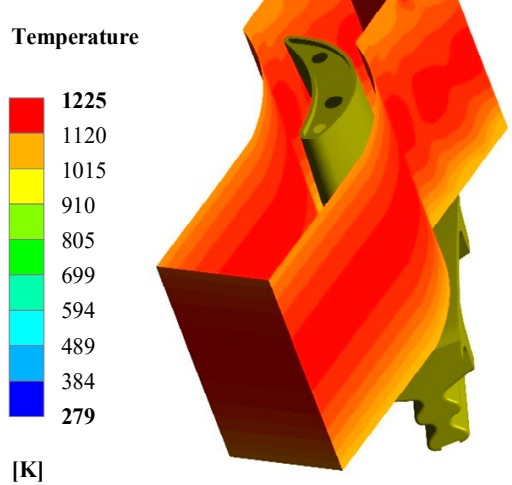

**Figure 5.** Temperature distribution of mainstream gas inlet.

It is necessary to specify that all theoretical studies on the behavior of turbine blades under thermo-mechanical fatigue were carried out at much higher parameters than those of the normal operation of the blades, i.e., the rotation speed was set to 36,000 rpm [25]. The cooling gas enters the cooling channel in a way perpendicular to the plane of the air inlet, and its inlet turbulence is set to 5%; the specific boundary condition-related parameters are shown in Table 3. The side of the lower edge plate of the blade is set as a fixed constraint. In order to simplify the calculation, it is assumed that the turbine blade endures three stages: take-off, cruising, and landing [26].

**Table 3.** Boundary parameters of the fluid domain under steady-state conditions.

| Parameters | Value | Units |
|:---|:---:|:---:|
| Total temperature at the inlet of the main gas flow domain | $T_{in}$ | K |
| Total pressure at the inlet of the main gas flow domain | 1.5 | MPa |
| Molar mass of gas | 28.29 | kg/kmol |
| Specific heat capacity of gas | $C_{pgas}$ | J/kg·K |
| Kinematical viscosity coefficient of gas | $1.831 \times 10^{-5}$ | kg/m·s |
| Thermal conductivity coefficients of gas | $2.61 \times 10^{-5}$ | W/m·K |
| Mass flow at the outlet of the main gas flow domain | 6.5632 | kg/s |
| Mass flow at the inlet of cooling airflow | 0.3321 | kg/s |
| Total temperature at the inlet of cooling airflow | 573.15 | K |
| Rotation speed | 36,000 | rpm |

According to the actual working conditions of the gas turbine and the results of Menter et al. [27] and Ho et al. [28], the SST k-ω model is more consistent with the experimental results in the fluid–solid coupling calculations; thus, it was chosen for numerical simulation. This calculation is considered to have converged when the residual values for all variables fall below $10^{-4}$ [11].

### 2.3. Thermal–Fluid–Solid Coupling Numerical Method

During the normal operation of the aero-engine, the blade is not allowed to have large deformation, so the unidirectional coupling method is used for calculation [12–14]. In this paper, the coupled thermal–flow–solid solution is divided into two steps. The flow and temperature fields of the blades and TBCs are first solved by the computational fluid dynamics (CFD) method. Then, the flow and temperature fields calculated in the first step are used as boundary conditions to solve the stress and strain fields of the blade and TBCs by the finite element method.

The flow and temperature fields of the blade are computed and analyzed using ANSYS Fluent 2020R2 commercial fluid dynamics solver software. The coupled solver is utilized to simulate the three-dimensional steady-state compressible viscous flow. The following time-averaged continuity equation, Navier–Stokes (NS) equation, and energy equation were solved as follows:

$$\frac{\partial(\rho u_j)}{\partial x_j} = 0$$

$$\frac{\partial(\rho u_i u_j)}{\partial x_j} = \frac{\partial p}{\partial x_i} + \frac{\partial}{\partial x_j}\left[(\mu + \mu_t)\left(\frac{\partial u_i}{\partial x_j} + \frac{\partial u_j}{\partial x_i}\right)\right] - \frac{2}{3}\frac{\partial}{\partial x_i}\left[(\mu + \mu_t)\left(\frac{\partial u_j}{\partial x_j}\right)\right] \quad (2)$$

$$\frac{\partial(\rho u_j h)}{\partial x_j} = \frac{\partial}{\partial x_j}\left(\lambda \frac{\partial T}{\partial x_j}\right) + \frac{\partial}{\partial x_j}(u_j \tau_{ij}) + S_{EF}$$

where $h$ is the total enthalpy of the fluid and $\lambda$ is the fluid heat transfer coefficient. $\partial(u_j \tau_{ij})/\partial x_j$ represents the part of the energy conversion from mechanical energy to thermal energy because of viscosity, which is called the dissipation function. $S_{EF}$ is the internal heat source of the fluid.

In order to close the NS equation, the equation of state for density and enthalpy needs to be added. The equation of state can be written as follows:

$$\rho = \frac{P}{R_0 T}$$
$$dh = c_p(T)dT \tag{3}$$

where $P$ is the absolute pressure and $R_0$ = 8.314 J/mol·K is the general gas constant.

The coupled method is used to solve heat transfer problems in the solid domain. Since there is no flow in the solid domain and only heat conduction exists, the heat conduction equation in the solid domain can be expressed as follows:

$$\frac{\partial}{\partial t}(\rho C_p T) = \nabla \cdot (\lambda \nabla T) + S_E \tag{4}$$

where $S_E$ is the internal energy source of the solid. For the thermal conductivity of the turbine blades, there is no heat or cold source inside the metal, so $S_E$ = 0.

Thermal stresses are caused by variations or non-uniform distribution of temperature across the blade, and the following equation calculates the modeled thermal stresses.

$$\sigma = E\alpha\Delta T \tag{5}$$

where $\sigma$ is the thermal stress, $E$ is the modulus of elasticity of the material, $\alpha$ is the coefficient of thermal expansion, and $\Delta T$ is the temperature change gradient.

### 2.4. Fatigue Life Analysis Method

The life of a part under variable cyclic loading is determined by a combination of different load levels and their number of cycles. Each level of loading and every cycle has an impact on part life [29]. Therefore, to analyze the fatigue life of a part under variable cyclic loading, it is necessary to first quantitatively evaluate the damage caused to the part by each cycle of different load levels. Palmgren and Miner successively proposed the linear cumulative damage theory of fatigue damage, which can quantitatively evaluate the contribution of different load levels to fatigue life [30].

The fatigue damage function $D(0, N) = 0$ satisfies the following conditions:

$$\begin{cases} D(0, N) = 0 \\ D(N, N) = D_{CR} \end{cases}$$
$$D = \sum_{i=1}^{p} \frac{n_i}{N_i} = D_{CR} \tag{6}$$

where $N_i$ is the fatigue damage produced when the external load is applied $n_i$ times and $D$ is the damage.

For several years, the widely used prediction models for blade low circumference fatigue prediction are the prediction model based on the Manson–Coffin theory and the linear cumulative damage theory [30], whose generalized formulas for fatigue life prediction are as follows:

$$\frac{\Delta\varepsilon_{eq}}{2} = \frac{\sigma'_f}{E}\left(2N_f\right)^b + \varepsilon'_f\left(2N_f\right)^e \tag{7}$$

where $\Delta\varepsilon_{eq}/2$ is the Mises equivalent strain amplitude; $\sigma'_f$ is the fatigue strength coefficient; $\varepsilon'_f$ is the fatigue plasticity coefficient; $b$ is the fatigue strength index; and $N_f$ is the number of low-week fatigue life cycles. Since the calculations in this paper only consider the effect of stress on the fatigue life of the TBCs and blade, and there is no centrifugal loading, the above equation does not require an average stress correction based on the centrifugal force of the blade.

## 3. Results and Discussion

### 3.1. TBCs and Blade Heat Transfer Analysis

Figure 6 shows the surface temperature distribution of the blade and TBCs at a cooling gas temperature of 573 K. As shown in Figure 6a, there is a localized high-temperature zone at the blade's leading edge, which can be seen in combination with Figure 5 since the middle of the leading edge is the first to be impacted by the high-temperature gas when the blade is in operation. In addition, the trailing edge of the blade also exists in the local high-temperature zone, which is due to the cooling gas undergoing multiple rotations and the effect of the disturbance column, the total pressure of the cooling gas in the trailing edge of the blade was greatly reduced, the cooling effect is weakened. The above blade temperature phenomena indicate that the blade temperature simulation results in this paper are reasonable. As shown in Figure 6c, the maximum temperature on the surface of the TC layer is 1153.12 K, and the maximum temperature on the surface of the blade is 1085.08 K. The thermal insulation effect of the TBC is obvious. The TBC's insulation effect at different locations of the blade needs to be further analyzed.

Figure 7 shows the temperature comparison between the blade with TBCs and the blade without TBCs at 50% height. As shown in Figure 7, the TBCs have a large difference in the thermal insulation effect at different locations of the blade, showing a better thermal insulation effect in the leading-edge region of the suction side of the blade, with a maximum temperature difference of 135.19 K. The TBCs show a poor thermal insulation effect in the region close to the trailing edge of the blade, with a minimum temperature difference of 14.13 K. The above results are in comparative agreement with the findings of Wei et al. [21] and Ziaei-Asl et al. [31] on the thermal insulation effect of TBCs, indicating that the thermal insulation capacity of TBCs is related to the geometry of the blade.

Figure 8 shows the temperature distribution of the blade cross-section at different heights. The high-temperature areas are concentrated in the leading edge at 10% and 50% height and the trailing edge at 90% height. The maximum temperature difference in the cross-section of different blade heights is reduced from 435 K (10% height) to 385 K (90% height). The temperature difference in the blade root section is large, and a thermal stress concentration zone is easily formed near the blade root.

### 3.2. Fatigue Life Analysis

Via the thermal–flow–solid coupling analysis in Section 3.1, the temperature fields of the blade and the TBCs are obtained. In this section, the temperature fields of the blade and TBCs are imported into the ANSYS finite element stress solver module as thermal load conditions for thermal stress calculations. Figure 9 shows the thermal stress distribution of the blade and TBCs, compared with the previous analysis, can be found that the highest temperature region does not correspond to the region of higher thermal stress. From Equation (5), the thermal stresses are caused by the variation or non-uniform distribution of temperature over the component, so the thermal stresses are higher in the region of the large temperature gradient.

As shown in Figure 9a, high thermal stress areas are located at the top of the blade at the air outlet (location 1), inside near the blade root (location 2), and at the bottom of the trailing edge of the blade (location 3). The high thermal stresses at location 1 and 2 are due to large temperature gradients, while the high thermal stress at location 3 is related to the discontinuity of the fillet geometry between the blade root and the platform, in addition to the large temperature gradient, which is in accordance with the findings of Cai et al. [11] on the blade stresses. Figure 9b,c show the thermal stress distributions of the BC and TC layers, respectively. The TC is a brittle material, and its failure is generally based on the maximum principal stress failure criterion [22]. Figure 10a shows the maximum principal stress distribution of the TC layer at different blade heights, and it can be observed that the TC shows the maximum tensile stress near the leading edge of the blade, which reaches 164.79 MPa. Figure 10b shows the stress distribution in the BC layer at different heights, and it is observed that the curves show less stress at the trailing edge of the blade.

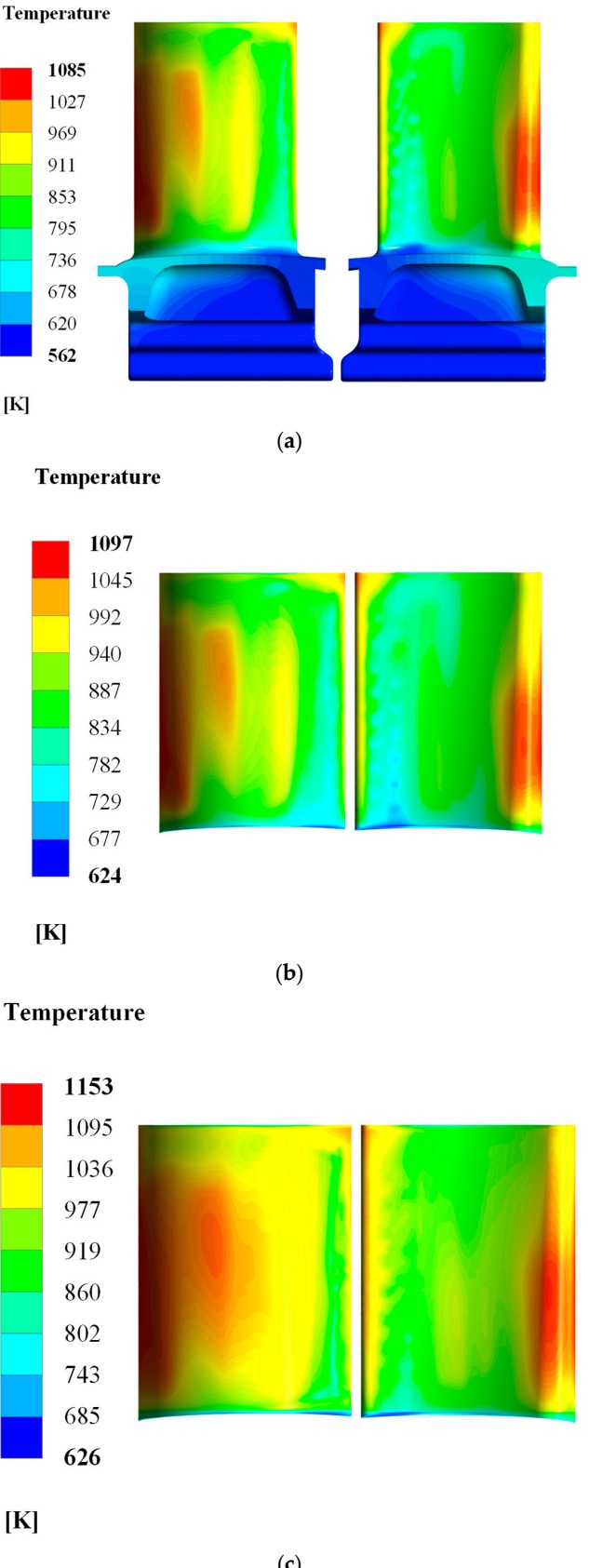

**Figure 6.** Temperature distribution of the blade with TBCs. (**a**) blade; (**b**) BC; (**c**) TC.

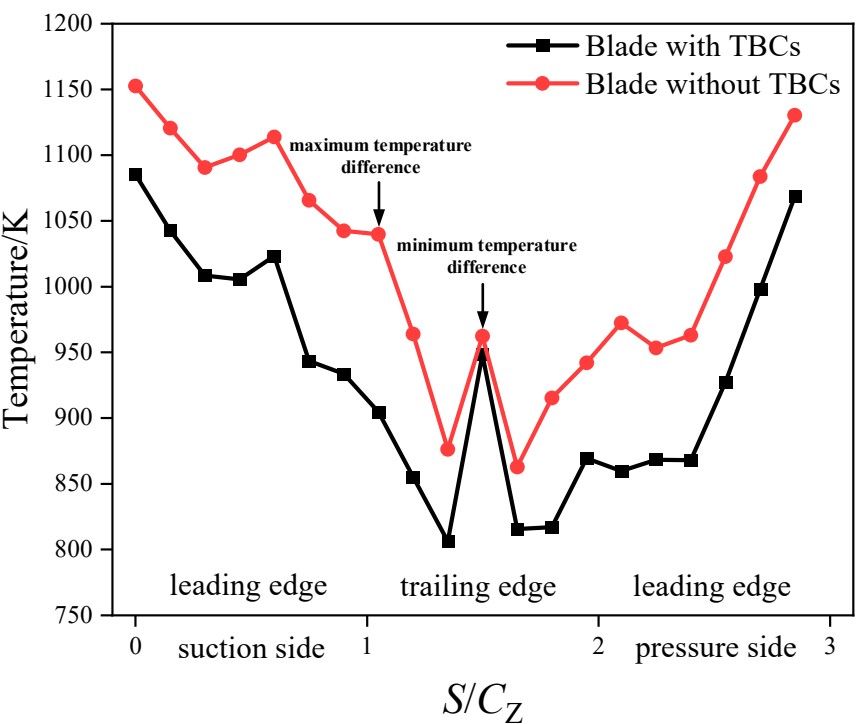

**Figure 7.** Temperature distribution of blade and TC at different positions of blade.

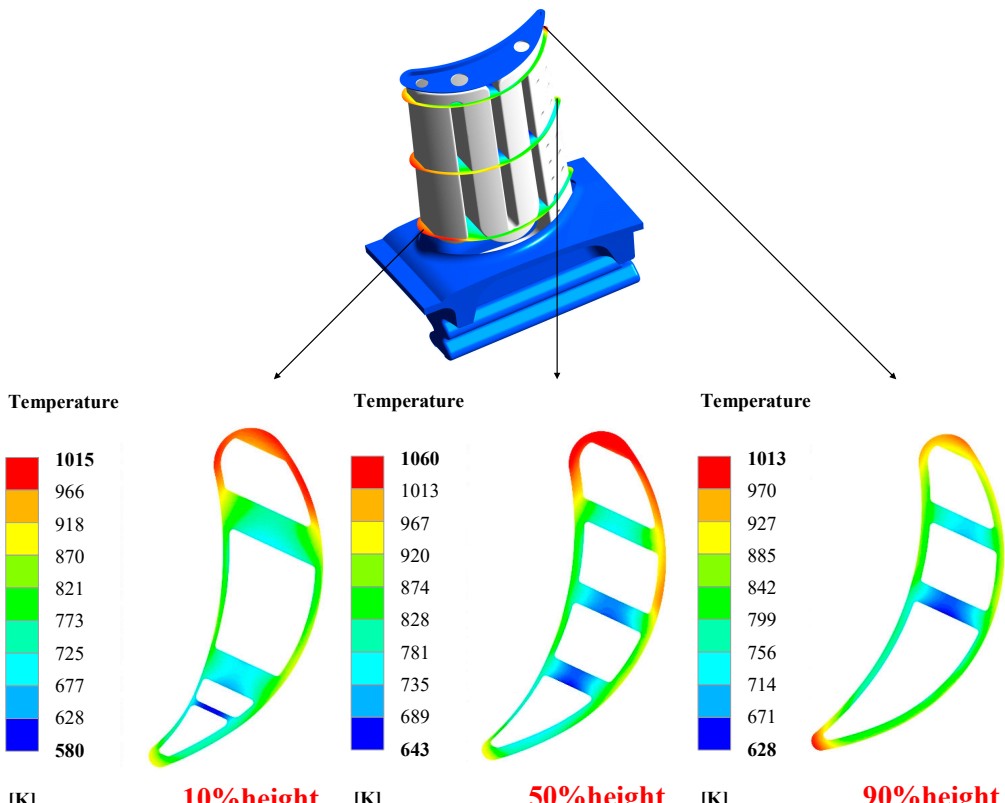

**Figure 8.** Temperature distribution of the blade cross-section at different heights.

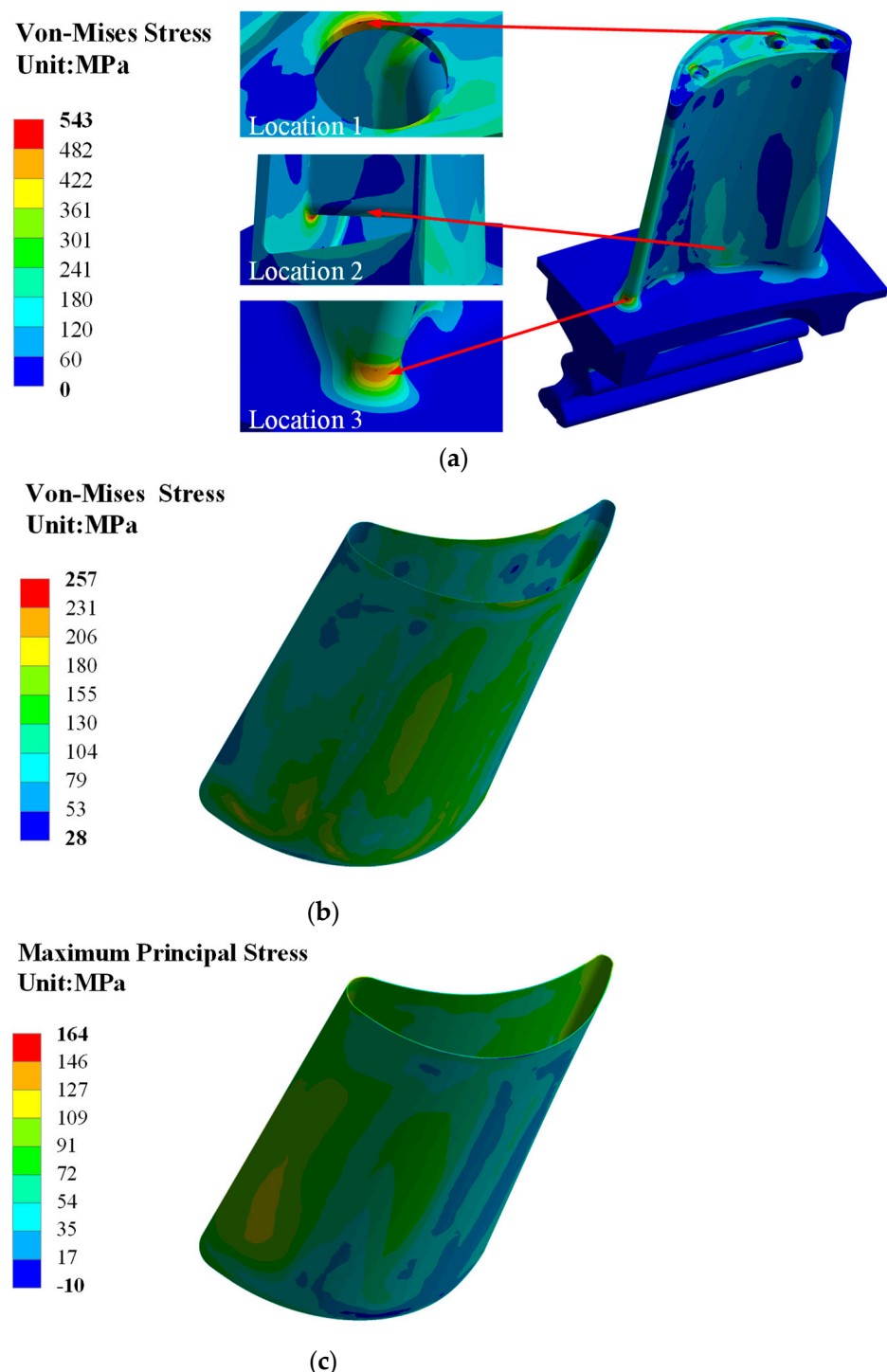

**Figure 9.** Stress distribution of blade with TBCs: (**a**) blade; (**b**) BC; (**c**) TC.

The resulting stress data are imported into Ncode 2020R2 for fatigue life analysis. Since the blades are not all pulsating cycles in the actual operating condition, Goodman's equation is used to convert all non-pulsating cycles into pulsating cycles when solving the fatigue life in Ncode. The solution procedure is set to be temperature-dependent and is based on the maximum stress criterion and a standard solver. Figure 11a,b show the fatigue life of the blade and the TBCs. It can be seen that the fatigue life of the blades and TBCs is relatively low in the region of high thermal stress. It is worth noting that the maximum stress point does not correspond to the lowest point in the life cycle. The most failure-prone locations are because they consistently deal with high stress levels or

large variations in stress amplitude, whereas the maximum stress point may not always maintain the maximum stress value. The minimum life of the blade is 7074 cycles, and the minimum life of the TBCs is 5348 cycles. The predicted hazardous areas are compared with the experimental results [32,33]. As can be seen in Figure 11b, the experimental TBCs shedding areas are relatively consistent with the predicted areas.

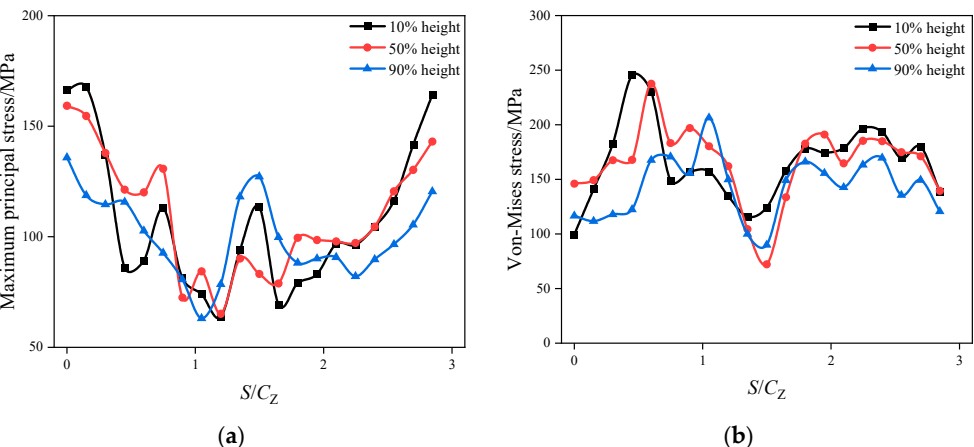

Figure 10. Stress changes in BC and TC at different heights: (a) TC; (b) BC.

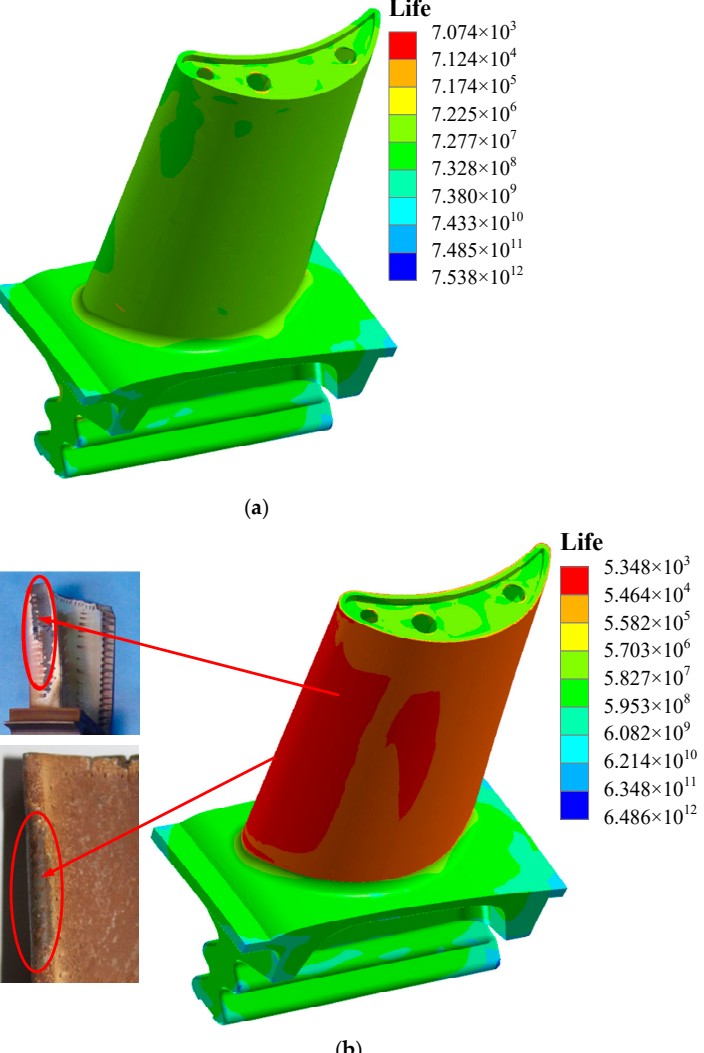

**Figure 11.** The fatigue life contours of blade and TBCs: (**a**) blade; (**b**) TBCs.

### 3.3. Influence of Cooling Gas Temperature on the Fatigue Life of Blade with TBCs

Larger temperature gradients create larger thermal stresses, which in turn affect the life of the blade and TBCs. Increasing the cooling gas temperature while the gas temperature remains unchanged can effectively reduce the temperature gradient of the blade substrate and coating, which is very helpful for the reduction in the overall stress and the improvement in life.

Firstly, the effect of different cooling gas temperatures on blade and TBC temperatures is investigated. As shown in Figure 12, the results indicate that the maximum and average temperatures of the blade and TBCs have increased to different degrees as the cooling gas temperature increases from 573 K to 1173 K. The cooling gas temperature has a large impact on the average temperature of each part, and the increase in cooling gas temperature leads to a substantial increase in the average temperature of the blade due to the direct contact between the cooling gas and the blade. However, the cooling gas temperature had a small effect on the maximum temperature of each part, with the smallest effect on the maximum temperature of the TC layer.

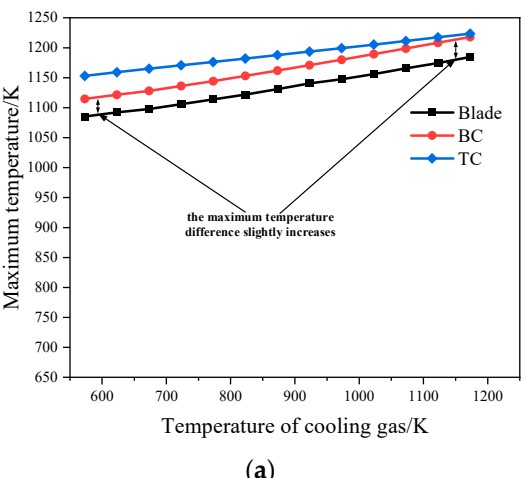

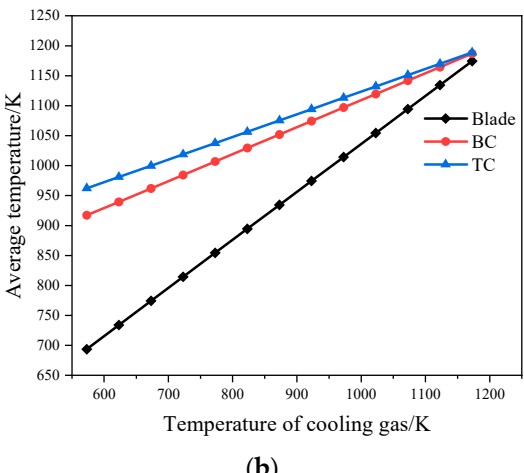

(**a**)  (**b**)

**Figure 12.** Temperature change in the blade with TBCs at different cooling gas temperatures: (**a**) maximum temperature; (**b**) average temperature.

Figure 13 shows the variation of the maximum stresses of the blade and TBCs with cooling gas temperature. It can be seen that the maximum stresses in both the blade and TBCs decrease to varying degrees as the cooling gas temperature increases, but the decreasing trend slows down. As the cooling gas temperature increased from 573 K to 973 K, the maximum blade stress decreased by 53.4%, and the maximum TBC stress decreased by 48.3%. The maximum stresses in the blade increase slightly after the cooling gas temperature exceeds 973 K. As shown in Figure 12a, it may be because when the cooling gas temperature exceeds 973 K, the increase amplitude in the maximum temperature of the blade is slightly reduced, and the difference in the maximum temperature with the BC layer is slightly increased.

Figures 14 and 15 show the fatigue life of the blade and TBCs at cooling gas temperatures of 573 K, 873 K, and 1173 K. It can be found via Figures 14 and 15 that changes in cooling gas temperature do not change the location of the hazardous area. Figure 16 shows the variation of the fatigue life of the blade and TBCs with changes in cooling gas temperature. The overall trend of blade and TBCs fatigue life is opposite to that of stress as the cooling gas temperature increases, and the stress reduction effectively improves service life. At a cooling gas temperature of 873 K, the minimum life of the blades and TBCs are maximized at 32,440 cycles and 12,770 cycles, respectively. When the cooling gas temperature is increased from 573 K to 873 K, the minimum life of the blades is increased by 358.5% and that of the TBCs by 138.7%. It is worth mentioning that the minimum fatigue life and maximum stress did not change the trend at the same cooling gas temperature.

After the cooling gas temperature is increased to 873 K, the material properties (modulus of elasticity, coefficient of thermal expansion, etc.) of the blade and TBCs gradually decrease at high temperatures due to the increase in the overall average temperature, resulting in a simultaneous decrease in the minimum life of the blade and TBCs in the presence of decreasing stresses. In summary, an appropriate increase in cooling gas temperatures can be effective in reducing stress and increasing fatigue life but will not change the failure hazard area.

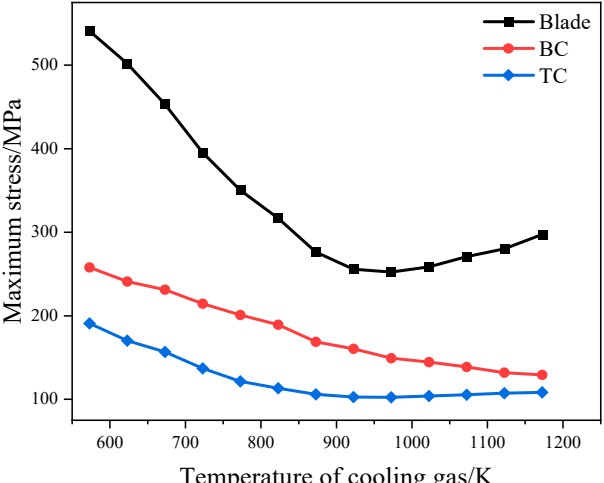

**Figure 13.** The maximum stress change in the blade with TBCs at different cooling gas temperatures.

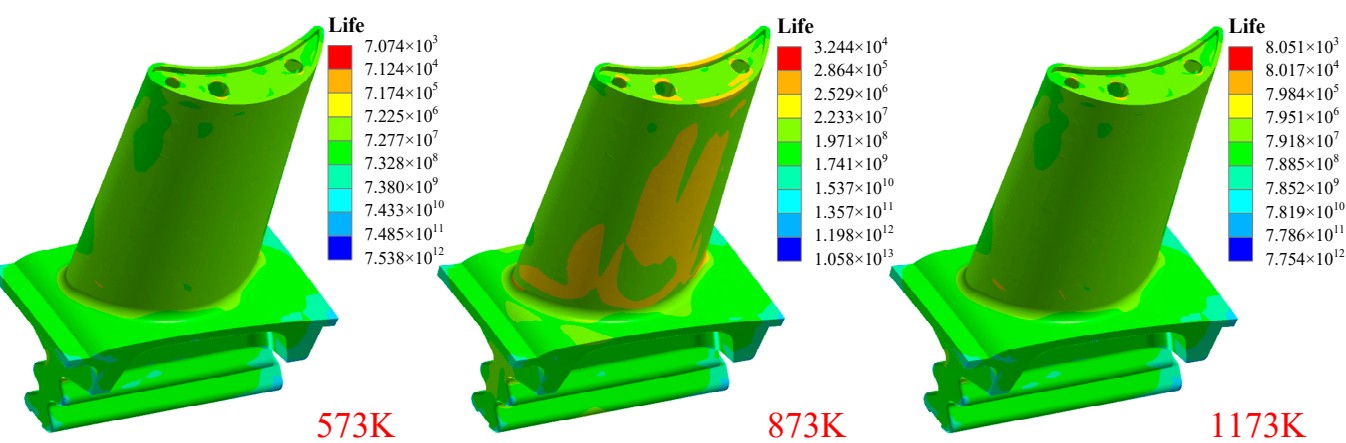

**Figure 14.** The fatigue life of the blade at cooling gas temperatures of 573 K, 873 K, and 1173 K.

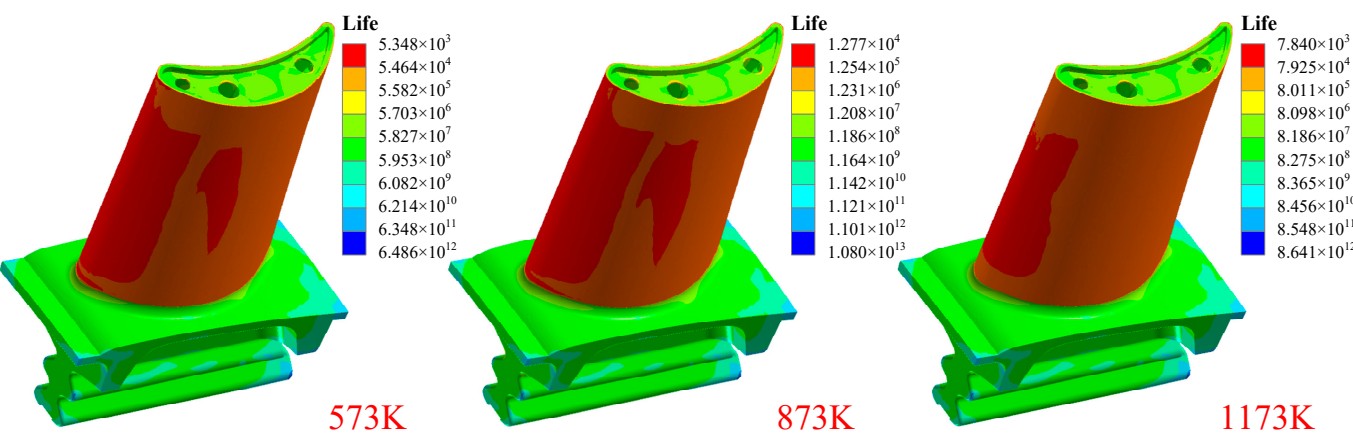

**Figure 15.** The fatigue life of TBCs at cooling gas temperatures of 573 K, 873 K, and 1173 K.

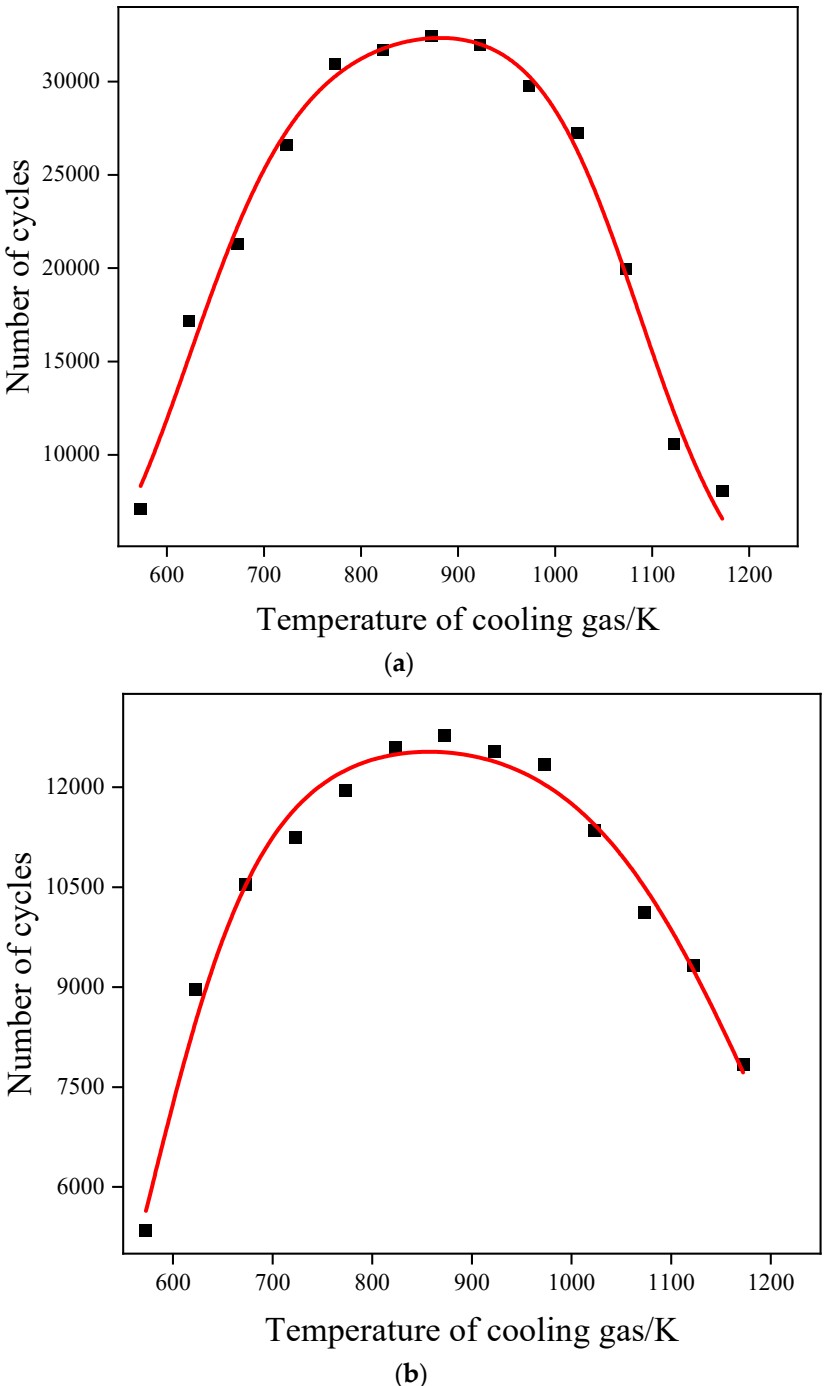

**Figure 16.** The fatigue life of blade and TBCs at different cooling gas temperatures: (**a**) blade; (**b**) TBCs.

## 4. Conclusions and Future Work

In this paper, the fatigue life of gas turbine blades with TBC was investigated. Considering the inhomogeneity of the high-temperature gas inlet temperature, the internal cooling gas temperature, and the periodicity of the external flow field, a finite element model of the gas turbine blade with TBCs was established to analyze the temperature field and stress field of the blade and the TBCs at different locations. Then, the life of the blade and the TBCs was predicted based on Ncode 2020R2. Finally, the effect of the internal cooling gas temperature on the stress and fatigue life of the blade and the TBCs was analyzed. The main conclusions are as follows:

(1) The thermal insulation effect of the TBCs is different in different positions of the blade, and it shows a better heat insulation effect in the leading-edge region of the suction side of the blade, with a maximum temperature difference of 135.19 K. In the region close to the trailing edge of the blade, the TBCs show a poorer thermal insulation effect, with a minimum temperature difference of 14.13 K. This provides data support for the future design of thermal insulation for blades with TBCs.

(2) The localized high thermal stresses in the blade and TBCs are related to the geometry of the blade in addition to the temperature gradient, and the fatigue life shows a significant negative correlation with the stresses. The fatigue life of the TBCs is lower than that of the blade, and the low-life region of the TBCs is located at the leading edge of the blade, which is consistent with the TBCs shedding region of the real blade and verifies the accuracy of the life prediction method in this paper. Providing new ideas for life prediction of blades with TBCs.

(3) The life of the blade and TBCs increases and then decreases with increasing cooling gas temperature, with the opposite trend for stress. When the cooling gas temperature was increased from 573 K to 973 K, the maximum stress in the blade decreased by 53.4%, and the maximum stress in the TBCs decreased by 48.3%. When the cooling gas temperature is increased from 573 K to 873 K, the minimum life of the blade increases by 358.5%, and the minimum life of TBCs increases by 138.7%. Appropriately increasing the cooling gas temperature can effectively reduce the localized high stress levels in the blades and TBCs while increasing the service life of the blades and TBCs. This finding can provide guidance for the design of long-life blades with TBCs.

The research work in this paper is a fundamental work which needs to be further improved to ensure that the blades with TBCs can operate for a long time under more severe environmental conditions. In future work, the effect of variable cycle cooling gas on the fatigue life of blades with TBCs will be investigated, such as the effect of cooling gas warming rate on the results. The optimal correspondence between high-temperature gas temperature and cooling gas temperature will also be investigated.

**Author Contributions:** Conceptualization, Y.C.; methodology, Y.C., Z.Z. and Y.A.; software, Z.Z. and P.G.; writing—original draft preparation, Y.C. and Z.Z.; writing—review and editing, Y.Y. and H.L. All authors have read and agreed to the published version of the manuscript.

**Funding:** This research was funded by the National Defense Technology Industry Innovation Foundation of China, grant number JJ202170301.

**Institutional Review Board Statement:** Not applicable.

**Informed Consent Statement:** Not applicable.

**Data Availability Statement:** Data are contained within the article.

**Acknowledgments:** The authors would like to thank the National Defense Technology Industry Innovation Foundation of China (No. JJ202170301) for funding.

**Conflicts of Interest:** The authors declare no conflict of interest.

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
