# Peer review of "Thermal–Fluid–Solid Coupling Analysis on the Effect of Cooling Gas Temperature on the Fatigue Life of Turbine Blades with TBCs"

_coatings, doi:10.3390/coatings13101795_

Round 1
Reviewer 1 Report
Overall a nice paper. I would just proofread and correct grammatical errors in some sections.
Has some grammatical errors that need to be corrected.
Author Response
Please see the atachment.

Reviewer 2 Report
No comments.
English is good.
Reviewer 3 Report
Kindly find my comments in the attached file.

Reviewer 4 Report
This is an interesting article on the effect of cooling gas temperature on the fatigue life of the blade with thermal barrier coating. This is an important issue from both a scientific and practical point of view. However, before allowing for publication, I would suggest making some minor improvements:
- Figure 2 is not precise. What are the diameters of the channels, the overall dimensions, the distances between the channels. This should be completed,
- table 1 - Error! Reference source not found,
- table 2 - the source of the data should be provided,
- whether the centrifugal force is taken into account,
- what state corresponds to the blade loads?
- whether any boundary conditions were used for the blade fixing ? Heat is also transferred to the vane,
- does this blade come from an actual engine or is it just an example model ?
- Figures 2 and 8 are confusing, as different channels are visible. I suggest making an additional figure with a cross-section of the blade,
- conclusions - what are the directions for further work ?
After completing the above mentioned comments, I recommend the paper for publication.
